# EMBEDDING MODELS THROUGH THE LENS OF STABLE COLORING

## ABSTRACT

Embedding-based approaches find the semantic meaning of tokens in structured data such as natural language, graphs, and even images. To a great degree, these approaches have developed independently in different domains. However, we find a common principle underlying these formulations, and it is rooted in solutions to the stable coloring problem in graphs (Weisfeiler-Lehman isomorphism test). For instance, we find links between stable coloring, distribution hypothesis in natural language processing, and non-local-means denoising algorithm in image signal processing. We even find that stable coloring has strong connections to a broad class of unsupervised embedding models which is surprising at first since stable coloring is generally applied for combinatorial problems. To establish this connection concretely we define a mathematical framework that defines continuous stable coloring on graphs and develops optimization problems to search for them. Grounded on this framework, we show that many algorithms ranging across different domains are, in fact, searching for continuous stable coloring solutions of an underlying graph corresponding to the domain. We show that popular and widely used embedding models such as Word2Vec, AWE, BERT, Node2Vec, and Vis-Transformer can be understood as instantiations of our general algorithm that solves the problem of continuous stable coloring. These instantiations offer useful insights into the workings of state-of-the-art models like BERT stimulating new research directions.

## 1 INTRODUCTION

Embedding models are ubiquitous in wide range of real-world applications such as information retrieval (Zuccon et al., 2015), natural language processing (NLP) (Mikolov et al., 2013a;b), graph classification (Grover & Leskovec, 2016; Hamilton et al., 2017) and many more. These models map categorical entities to continuous dense representations (typically in $\mathbf{R}^d$) which provide a continuous measure of semantic similarity across categorical entities. Nowadays, there is a heavy dependence on unsupervised pre-trained embedding models across domains like Transformers in NLP (Devlin et al., 2019), Visual Transformers (ViT) in Computer Vision (Dosovitskiy et al., 2020), Graph Neural Networks (Hamilton et al., 2017; Xu et al., 2019b) since they learn rich semantic representations of entities from massive amounts of unlabelled data. With little finetuning, these models achieve state-of-the-art results on most of the supervised downstream tasks like sentiment analysis (Xu et al., 2019a), object detection (Beal et al., 2020), and graph classification (Xu et al., 2019b).

Historically, embedding models were developed almost independently across structured domains such as NLP, graphs, images, and so on. These algorithms use the neighborhood structure around an entity to obtain the embedding for the entity. Interestingly, a popular hypothesis in NLP - Distributional Hypothesis states that the "meaning" of the word is determined by its context (neighbors) (Harris, 1954; Sahlgren, 2008). This hypothesis forms the basis of most unsupervised embedding learning models in NLP (Mikolov et al., 2013a;b; Pennington et al., 2014b; Bojanowski et al., 2017; Sonkar et al., 2020). Similarly, non-local-means, a denoising algorithm in signal processing, tries to find pixels that should be the same based on the similarity of its neighborhood structure (patch of the image around the pixel in this case) (Awate & Whitaker, 2006; Buades et al., 2005). Even Graph neural network (GNN) architectures ensure that information of the surrounding neighbors is systematically incorporated in the embedding of a node, even in supervised settings (Hamilton et al., 2017; Maron et al., 2019; Xu et al., 2019b). Thus somehow various communities working across

diverse domains have narrowed down on an entity's neighborhood structure to define the entity's meaning. In this paper, we try to establish this common principle on mathematically robust grounds.

Structured domains can be easily represented as graphs with relations between entities as edges in the graphs. For example in NLP words can be treated as nodes of some graph, and co-occurrence relation between words can be represented as an edge. As mentioned before that graph embedding architectures capture the topological structure around the node in the node embeddings, and if structured domains can be represented as graphs, this raises the question do embedding models from structured domains like NLP and Vision also operate on some domain induced graph and capture neighborhood structural properties in their entity embeddings since we have seen that embedding models across these domains tend to capture "neighborhood" information?

To answer this question, we dive into combinatorial graph theory to understand how to define the notion of structural equivalence a.k.a. isomorphic structures. Weisfeiler-Lehman (1-WL) algorithm (or color refinement algorithm) is the most popular heuristic used to identify graph isomorphism (Weisfeiler & Leman, 1968) and can distinguish a broad class of graphs (Babai & Kucera, 1979). The fixed point solution of 1-WL is called a *stable-coloring* and has the property that any two nodes with the same color have the same multi-set of colors in their neighborhood. In essence it means if two nodes have the same color, the graph looks structurally identical from these nodes.

Finally to answer the question if embedding models from structured domains capture the domain induced graph's topological properties in their entity embeddings, one can find links between the mechanics of these models and stable coloring / 1-WL algorithm. In this paper, we establish this connection by providing a general framework linking existing algorithms to stable coloring. We propose a more flexible version of stable coloring (SC) called continuous stable coloring (CSC ) — a strict generalization of SC. CSC states that the similarity of their neighborhoods determines the similarity of two nodes. Based on this notion, we define a series of optimization problems to solve the problem of CSC . We show that various algorithms in NLP like word2vec, AWE, BERT (Devlin et al., 2019), images processing like Visual Transformer (Dosovitskiy et al., 2020), graphs like Node2Vec (Grover & Leskovec, 2016), etc. are essentially solving different instantiations of this common optimization problem.

Current research already establishes the link between the 1-WL algorithm (Grohe, 2020; Morris et al., 2021; Shervashidze et al., 2011; Morris et al., 2017) and GNN architectures, which has sparked a new line of research in improving GNN architectures (Hamilton et al., 2017; Xu et al., 2019b; Maron et al., 2019; Morris et al., 2020a;b). We hope that the new link we establish between stable coloring and unsupervised embedding algorithms will also stimulate new exciting research in embeddings for other structured domains of NLP and Vision.

## 2 BACKGROUND

In this section, we define a stable colored graph, provide an outline for 1-WL (Weisfeiler-Lehman) graph isomorphism test and General Aggregate and Update (GAU ) for Graph Neural Networks (GNNs). We also discuss how domains of NLP and images can be seen as graphs.

### 2.1 STABLE COLORING

Let a coloring function $\mathcal{C}$ be an overloaded function defined on the vertices as well on set of vertices of $\mathcal{G} = (\mathcal{V}, \mathcal{E})$ , i.e., $\mathcal{C} : \mathcal{V} \to \mathbf{N}$ and $\mathcal{C} : 2^{\mathcal{V}} \to \{\{\mathbf{N}\}\}$ where $\mathbf{N}$ is a set of natural numbers representing colors, and $\{\{.\}\}$ is a multiset with overloading defined as $\mathcal{C}(A) = \{\{\mathcal{C}(v) | v \in A \subset \mathcal{V}\}\}$. We denote neighborhood of a node $u \in \mathcal{V}$ as $\mathcal{N}(v) = \{u | (v, u) \in \mathcal{E}\}$.

**Definition 2.1** (Stable Coloring). An undirected graph $\mathcal{G} = (\mathcal{V}, \mathcal{E})$ is stable colored w.r.t coloring function $\mathcal{C}$ if it holds that $\mathcal{C}(u) = \mathcal{C}(v)$ if and only if $\mathcal{C}(\mathcal{N}(u)) = \mathcal{C}(\mathcal{N}(v))$.

We can extend the above definition to directed graphs and graphs with labels. A directed graph $\mathcal{G} = (\mathcal{V}, \mathcal{E})$ is stable colored w.r.t a coloring function $\mathcal{C} : \mathcal{V} \to \mathbf{N}$ if it holds that $\mathcal{C}(u) = \mathcal{C}(v)$ if and only if $\mathcal{C}(\mathcal{N}_{in}(u)) = \mathcal{C}(\mathcal{N}_{in}(v))$ and $\mathcal{C}(\mathcal{N}_{out}(u)) = \mathcal{C}(\mathcal{N}_{out}(v))$ where $N_{in}(u) = \{w | (w, u) \in \mathcal{E}\}$ and $N_{out}(u) = \{w | (u, w) \in \mathcal{E}\}$. Likewise, an edge-labelled undirected graph $\mathcal{G} = (\mathcal{V}, \mathcal{E})$ is stable colored w.r.t a coloring function $\mathcal{C} : \mathcal{V} \to \mathbf{N}$ if it holds that $\mathcal{C}(u) = \mathcal{C}(v)$ if and only if $\forall l, \mathcal{C}(\mathcal{N}_l(u)) = \mathcal{C}(\mathcal{N}_l(v))$ where $N_l(u) = \{w | (u, w) \in \mathcal{E}_l\}$ where $\mathcal{E}_l \subset \mathcal{E}$ for an edge label $l$.

We also define a $weak$-stable coloring for a graph $\mathcal{G} = (\mathcal{V}, \mathcal{E})$. An undirected graph $\mathcal{G} = (\mathcal{V}, \mathcal{E})$ is $weak$-stable colored w.r.t coloring function $\mathcal{C}$ if it holds that $\mathcal{C}(u) = \mathcal{C}(v)$ if $\mathcal{C}(\mathcal{N}(u)) = \mathcal{C}(\mathcal{N}(v))$ where $\mathcal{N}(u) = \{w|(u, w) \in \mathcal{E}\}$.

## 2.2  1-WL ALGORITHM AND GENERAL AGGREGATE AND UPDATE FRAMEWORK (GAU )

**1-WL algorithm:** 1-WL is an iterative algorithm to achieve a stable coloring $\mathcal{C}$ for $\mathcal{G} = (\mathcal{V}, \mathcal{E})$. Let $\mathcal{C}_i$ denote the coloring at iteration $i$. It starts with a coloring scheme $\mathcal{C}_0$ such that $\mathcal{C}_0(v)$ is same for all $v \in \mathcal{V}$. In each iteration, it assigns a different color to $u$ and $v$ if $\mathcal{C}_i(\mathcal{N}(u)) \neq \mathcal{C}_i(\mathcal{N}(u))$ until a stable coloring $\mathcal{C}$ is reached.

**General Aggregate and Update Framework (GAU )** For a general multi-layer GNN, the General Aggregate and Update framework to compute node/vertex embeddings (corresponding to colors) of $\mathcal{G} = (\mathcal{V}, \mathcal{E})$ is given iteratively by:

$$\mathbf{E}^k(u) = f_{update}^{(k)}\left(\mathbf{E}^{k-1}(u), f_{agg}^{(k)}\left(\mathbf{E}^{k-1}(v) : v \in \{\{\mathcal{N}(u)\}\}\right)\right), \tag{1}$$

where $f_{agg}^{(k)}(.)$ and $f_{update}^{(k)}(.)$ map vertex multiset embeddings to a metric space such as $R^d$.

In GraphSage (Hamilton et al., 2017), authors showed that the iterative procedure in 1-WL algorithm is analogous to General Aggregate and Update procedure in GNNs. This connection has lead to the research direction where 1-WL is being used as a standard to which GNN architectures are being compared to (Xu et al., 2019b; Maron et al., 2019; Morris et al., 2020b). Xu et al. (2019b) prove that GAU is as powerful as 1-WL if the functions $f_{update}$ and $f_{agg}$ are injective.

Various domains such as NLP, Images and Graphs can be viewed as a graph on their elementary tokens. How we construct graphs is explained in section 6.1 and examples are given in appendix B.

## 3  RELATED WORK

Word embeddings have been popular in NLP since decades (Deerwester et al., 1989; Morin & Bengio, 2005; Mikolov et al., 2013b; Bojanowski et al., 2017). A lot of work has been done to understand the mathematical underpinning of these models, for instance, relation of embedding models to co-occurrence statistics (Levy & Goldberg, 2014; Hashimoto et al., 2016; Allen et al., 2019). Study of empirical properties of these embedding models (e.g. analogies) has also attracted theoretical research (Allen & Hospedales, 2019; Ethayarajh et al., 2019).

Recently, graph community has also seen a surge in learning node and graph embeddings. The notion of capturing the structural neighborhood around a node inside the node embedding has been the driving principle of these semi-supervised node embedding algorithms like node2vec (Grover & Leskovec, 2016), and GraphSAGE (Hamilton et al., 2017). Hamilton et al. (2017) pointed out that their GraphSAGE node embedding algorithm mimics the aggregate and update procedure of 1-WL algorithm. Subsequently, these neighborhood informative node embeddings found their applications in constructing graph embeddings, and thereby used for classification of structurally equivalent or isomorphic graphs (Chen et al., 2019). Xu et al. (2019b) in their GIN (Graph Isomorphism Network) model modified the aggregate procedure of GraphSAGE to construct graph embedders which were provably as powerful as 1-WL algorithm in distinguishing non-isomorphic graphs. This redirected the research into designing more powerful variants of graph embedders like PPGN (Maron et al., 2019) and $k$-GNN (Morris et al., 2019) which were provably as powerful as 3-WL and $k$-WL test respectively. Grohe (2020) discusses about these phenomenal works in increasing the expressivity of graph embeddings for supervised graph classification and regression tasks. While this line of research focuses on improving expressivity and generalizability of Graph networks based on its connection to 1-WL, we explore and formalize the unsupervised algorithms under the light of stable coloring / 1-WL algorithm and show that all the current models stem from the common principle that a tokens meaning is derived from its neighbours.

The rest of the paper is organised as follows. We begin with a discussion of connections between discrete stable coloring and non-local means algorithm in Image processing and distributional hypothesis in NLP. We then define a continuous version of SC and develop optimization problems to solve for CSC in section 5. In section 6 we show how current state-of-the art embedding models are solving the CSC problem in disguise.

## 4 Algorithms with Roots in Discrete Stable Coloring

Distributional hypothesis in NLP states that words that occur in similar contexts are semantically similar. Analogously, the non-local means image denoising algorithm in CV literature assigns similar intensity values to pixels that have similar patches surrounding them. The essence of both ideas is that the value of an entity is determined by its neighborhood. In this section, we connect these two ideas to the idea of stable coloring which states that if two nodes in a stable colored graph have same colored neighborhood, they have the same color. The proofs of the following theorems are provided in appendix.

**Theorem 4.1.** (Distributional Hypothesis(DH) in NLP encodes SC) Let $\{w_1, w_2, .., w_n\}$ be words in vocabulary, and $\mathbb{M} \in \mathbb{R}^{n \times n}$ is a co-occurrence matrix with entries $e_{ij} \in \mathbf{N}$ containing the number of times $w_i$ co-occurs with $w_j$ within a fixed context window. Let there be a function $f : w \to \mathbf{N}$ which takes a word and maps it to a color $c \in \mathbf{N}$ (assigns meaning in accordance with Distributional Hypothesis), such that $f(w_i) = f(w_j)$ only if row $i$ is same as row $j$ in matrix $\mathbb{M}$. Construct a graph $G_{DH}$ with words as nodes and edge labels given by $\mathbb{M}$. Then, function $f$ defines a stable coloring on graph $G_{DH}$.

DH defines words to be semantically same if they are substitutions of each other. In a true graph ( not the one created from samples) their co-occurrence frequencies with other words is exactly the same. Equating color assigned by $f$ to represent meaning of a word as per DH, it is easy to see that words that end with same color are semantically same.

**Theorem 4.2.** Consider a discrete signal $y(x)$ sampled at $n$ points $x_i$ $(i = 1, ..., n)$, and let the sequence $\boldsymbol{p}(x_i) = (y(x_{i-t}), .., y(x_{i-1}), y(x_{i+1}), .., y(x_{i+t}))$ be a patch of neighborhood values around each $x_i$ for some context window length $t$. NLM denoises the signal $y(x_i)$, $i = 1, ..., n$ with iterative updates. The fixed point denoised version of the signal $y_d$ can be written as follows:

$$y_d(x_i) = \frac{1}{D(x_i)} \sum_{j=1}^{n} K(\boldsymbol{p}_d(x_i), \boldsymbol{p}_d(x_j)) y_d(x_j), \tag{2}$$

where $D(x_i) = \sum_{j=1}^{n} K(y_d(x_i), y_d(x_j))$ and K is an arbitrary kernel function. Let the graph $\mathcal{G}_{nlm} = (\mathcal{V}_{nlm}, \mathcal{E}_{nlm})$ where each $x_i$ is a node $u_i$ in $\mathcal{V}_{nlm}$ and each pair $(u_i, u_j)$ with $|i - j| \leq t$ is represented as a directed edge in $\mathcal{E}_{nlm}$ with label $(i - j)$. Then the fixed point solution of NLM with Kronecker delta kernel $K_\delta$, $y_d : \mathbf{N} \to R$ defines a *weak*-stable coloring over the graph $\mathcal{G}_{nlm}$.

An image is an example of a discrete signal with pixel intensities as signal values. With Kronecker delta kernel, NLM terminates with the intensity value of $y_d(x_i) = y_d(x_j)$ only if $\boldsymbol{p}_d(x_i) = \boldsymbol{p}_d(x_j)$. Assigning color to node $n_i$ as $x_i$'s final denoised intensity value, one can observe that NLM only terminates when a $weak$-stable colored graph is defined over the final image. Note that if the pixel values have high variety, then the probability to get a stable-colored graph is high.

From the above two examples, it is interesting to observe that a stable colored graph $G$ emerges from the underlying principles used by two different domains. Both in DH and NLM, the words/pixels that end up with the same color have same colored neighborhood around them. In order to find more connections to stable coloring, especially that of embedding models, we need a continuous representation of color and hence in next section we start building on a novel notion of color.

## 5 General Aggregate Learning framework for CSC

As shown in the earlier sections, even discrete stable coloring can be connected to various concepts in NLP and image processing. We find even more deep rooted links between stable coloring and unsupervised learning algorithms in structured domains. In fact, we can view embeddings of each node as a continuous 'color' assigned to each node. In order to show these links, we first need to generalize the idea of SC to a setting where we can talk about continuously comparable colors. This section is organised as follows. We first define a CSC problem analogous to SC . We show that this is a strict generalization of SC (discrete) problem. Then, inspired by the GAU for stable coloring, we propose a series of learning problems having GAU at their core (called General Aggregate Learning , GAL in short) . We end this section with a generalized algorithm whose parameters, as we show in section 6, can be initialized in various ways to obtain algorithms in varied domains that have been a de facto standard in those domains since a long time.

## 5.1 Continuous Stable Coloring (CSC )

Let us consider graph $\mathcal{G} = (\mathcal{V}, \mathcal{E})$ representing the structured domain under consideration. In SC , we assign categorical colors to the nodes of the graph. However, for most analytical tasks, including machine learning, we need to assign continuous labels (or embedding) to the nodes. Hence, we introduce the problem of CSC . Let the domain of colors assigned be a metric space $\mathbf{L}$ (eg. $\mathbf{R}^d$) associated with a distance metric $\mathcal{D}$. There are various ways to define similarity metrics in literature based on $\mathcal{D}$. For the sake of discussion in this section, we would use a simple definition of $\mathcal{S}(x, y) = e^{-\mathcal{D}(x,y)}$. While the theorems with other definitions will change in appearance, they will still maintain the spirit of analysis. We denote the neighbourhood $\mathcal{N}(u) = \{v | (u, v) \in \mathcal{E}\}$. Let $\mathcal{C} : \mathcal{V} \to \mathbf{L}$ denote the coloring of nodes. We overload the function $\mathcal{C}$, $\mathcal{C} : 2^{\mathcal{V}} \to \mathbf{N}^{\mathbf{L}}$, to operate on subset of nodes as given in the following equation. We use the notation $\mathbf{N}^{\mathbf{L}}$ to denote all multi-subsets of $\mathbf{L}$.

$$\mathcal{C}(V) = \{\{\mathcal{C}(v) | v \in V \subseteq \mathcal{V}\}\} \tag{3}$$

We define $\mathcal{S}_{\mathcal{N}} : \mathbf{N}^{\mathbf{L}} \times \mathbf{N}^{\mathbf{L}} \to \mathbf{R}$ as the similarity metric over the multi-subsets of $\mathbf{L}$ via the same similarity $\mathcal{S}$ over $\mathbf{L}$ and a permutation invariant and injective aggregator function $f_{agg} : N^{\mathbf{L}} \to \mathbf{L}$ as

$$\mathcal{S}_{\mathcal{N}}(A, B) = \mathcal{S}(f_{agg}(A), f_{agg}(B)) \quad \text{where, } A, B \in \mathbf{N}^{\mathbf{L}}. \tag{4}$$

We refer to this embedding as the continuous color under CSC . We use this terminology interchangeably as is best for the context. We denote nodes by small case letters $(u, v, ...)$ and the subset/multi-subsets of nodes by upper case letters $(A, B, ..)$ etc. Let us now take a look at the continuous stable coloring formulation (CSC ).

**Definition 5.1.** (Continuous Stable Coloring $(\mathbf{L}, \mathcal{S}, f_{agg})$). The coloring $\mathcal{C} : \mathcal{V} \to \mathbf{L}$ of nodes in graph $\mathcal{G} = (\mathcal{V}, \mathcal{E})$ is called continuous stable coloring parameterised by similarity kernel $\mathcal{S} : \mathbf{L} \times \mathbf{L} \to R$ for some metric space $\mathbf{L}$ and an injective aggregator function $f_{agg} : \mathbf{N}^{\mathbf{L}} \to \mathbf{L}$ if the following holds:

$$\mathcal{S}(\mathcal{C}(u), \mathcal{C}(v)) = \mathcal{S}_{\mathcal{N}}(\mathcal{C}(\mathcal{N}(u)), \mathcal{C}(\mathcal{N}(v))), \quad \forall u, v \in \mathcal{V}. \tag{5}$$

In most applications, we look at $\mathbf{L} = R^d$ for some $d > 0$. Essentially, CSC states that the similarity between embeddings of two nodes should be equal to the similarity between the neighborhoods of the two nodes. Thus CSC relaxes the Kronecker delta function of comparison over categories in SC to a general similarity metric over $\mathbf{L}$. In fact CSC is a strict generalization of SC which we show in the next theorem.

**Theorem 5.1.** (SC is a special case of CSC ). Stable coloring (discrete) problem is an instance of continuous stable coloring problem with $\mathbf{L} = \mathbf{N}$, $\mathcal{S}(i, j) = \mathbb{1}(i = j)$ and $\mathcal{S}_{\mathcal{N}}(s_1, s_2) = \mathbb{1}(s_1 = s_2)$ where $i, j \in \mathbf{N}$ and $s_1, s_2 \in \mathbf{N}^{\mathbf{N}}$. In this case $f_{agg} : \mathbf{N}^{\mathbf{N}} \to \mathbf{N}$ function is an injective hash function which maps multi-subsets of $\mathbf{N}$ to $\mathbf{N}$.

It is easy to verify the validity of this theorem by using correct values for $\mathbf{L}$, $\mathcal{S}$ and $f_{agg}$ as mentioned in the theorem. Next, we define a learning problem GAL which solves the CSC problem.

## 5.2 General Aggregate Learning (GAL )

Consider $\mathcal{G} = (\mathcal{V}, \mathcal{E})$ representing the structured domain under consideration. In (Xu et al., 2019b), authors showed that GAU framework (equation 1) with an injective $f_{merge}$ and $f_{agg}$ operations is equivalent to 1-WL which solves SC problem. We define a solution to the CSC problem that is similar in spirit of GAU . However, instead of providing an iterative algorithm like GAU , we pose an optimization objective which, in essence, *learns* the stable solution directly.

First, we define the notations. Recall that the task is to assign continuous labels to graph nodes in metric space $\mathbf{L}$ with distance metric $\mathcal{D}$ and the similarity metric is given as $\mathcal{S}(x, y) = e^{-\mathcal{D}(x,y)}$. Notation for the embedding matrix $\mathbf{E}$ is defined in the same way as that of a coloring function $\mathcal{C}$, such that $\mathbf{E} : \mathcal{V} \to \mathbf{L}$ and $\mathbf{E} : 2^{\mathcal{V}} \to \{\{\mathbf{L}\}\}$:

$$\mathbf{E} : \text{Embedding matrix}, \quad \mathbf{E}(u) : \text{embedding of node } u \in \mathcal{V}, \text{and}$$
$$\mathbf{E}(V) = \{\{\mathbf{E}(v) | v \in V \subseteq \mathcal{V}\}\}. \tag{6}$$

$\mathbf{E}$ stores the color assignments of all nodes in $\mathcal{V}$ and is learned in our setting. Let us look at our first optimization objective which follows naturally from the definition of CSC .

**Definition 5.2.** (Global GAL formulation) Let matrix $\mathbf{E} \in R^{|\mathcal{V}| \times d}$ store embeddings (or colors) of nodes in graph $\mathcal{G} = (\mathcal{V}, \mathcal{E})$. Then the optimization objective of Global GAL (G-GAL) parameterized by an injective function $f_{agg} : \mathbf{N}^{\mathbf{L}} \rightarrow \mathbf{L}$ (where $\mathbf{L} = R^d$) is to learn an embedding matrix $\mathbf{E}$ that minimizes the following

$$\mathbf{E} = \arg\min_{E} \sum_{u,v \in \mathcal{V}} \text{abs}\Big( -\ln \mathcal{S}\big(E(u), E(v)\big) + \ln \mathcal{S}\big(f_{agg}(E(\mathcal{N}(u))), f_{agg}(E(\mathcal{N}(v)))\big)\Big) \quad (7)$$

The function $f_{agg}$ can be as simple as a sum operation or as complex as a neural network architecture with learnable parameters. The global formulation follows naturally from the definition of CSC $(R^d, \mathcal{S}, f_{agg})$ problem. Note that we project neighborhood embeddings( or coloring) into the same space as node embeddings. This will be important for subsequent formulations.

> **Research Question: Can an effective learning strategy be formulated for minimizing G-GAL loss?** An effective algorithm for G-GAL formulation can stimulate further research. To the best of our knowledge, we do not know of an algorithm in any domain which uses this formulation. We think a possible reason why it is difficult to learn is that in most applications, we only have access to subgraph samples of the true underlying graph. Working with a sample of neighbourhood $\hat{\mathcal{N}}(u)$ instead of the complete neighborhood implies we are estimating $\hat{f}_{agg}(E(\mathcal{N}(u)))$. The errors in $\mathcal{S}(\hat{f}_{agg}(E(\mathcal{N}(u))), \hat{f}_{agg}(E(\mathcal{N}(v))))$ increase super-linearly with error in $\hat{f}_{agg}(E(\mathcal{N}(u)))$. Nonetheless, this formulation can be of independent interest.

The above mentioned issue of multiplying noisy neighborhood estimates motivates us to find a simpler learning problem. We formulate a different learning problem and show in theorem 5.2 that this simpler problem, which we call Node-local GAL (L-GAL) problem also solves G-GAL problem.

**Definition 5.3.** (L-GAL formulation). Let matrix $\mathbf{E} \in R^{|\mathcal{V}| \times d}$ store embeddings (or colors) of nodes in graph $\mathcal{G} = (\mathcal{V}, \mathcal{E})$. Then the optimization objective of L-GAL parameterized by an injective function $f_{agg} : \mathbf{N}^{\mathbf{L}} \rightarrow \mathbf{L}$ (where $\mathbf{L} = R^d$) is to learn an embedding matrix $\mathbf{E}$ that minimizes the following

$$\mathbf{E} = \arg\min_{E} \sum_{v \in \mathcal{V}} -\ln \mathcal{S}(E(v), f_{agg}(E(\mathcal{N}(v)))). \quad (8)$$

In the above L-GAL formulation, we uncoupled the G-GAL formulation and thus eliminated the issue of multiplying noisy neighborhood estimates. However, we would like to emphasize that the solution to L-GAL is also a good solution to G-GAL. We quantify this relation in the next theorem.

**Theorem 5.2.** (L-GAL solves G-GAL). Let the solution $\mathbf{E}$ to L-GAL upper-bounds each term in the summation of loss in equation 8 by some $\epsilon > 0$; thus it upper-bounds total loss by $|\mathcal{V}|\epsilon$. Then the same solution matrix $\mathbf{E}$ is a solution to G-GAL with each term in summation upper-bounded by $2\epsilon$ and thus upper-bounding the total loss by $|\mathcal{V}|^2\epsilon$.

The proof of theorem 5.2 can be found in appendix. Uncoupling works because if you alternately bring $\mathbf{E}(u)$ and $\mathbf{E}(v)$ closer to $f_{agg}(\mathbf{E}(\mathcal{N}(u)))$ and $f_{agg}(\mathbf{E}(\mathcal{N}(v)))$ respectively, it forces the distances between $(\mathbf{E}(u), \mathbf{E}(v))$ and $(f_{agg}(\mathbf{E}(\mathcal{N}(u))), f_{agg}(\mathbf{E}(\mathcal{N}(v))))$ to be nearly equal.

## 5.3 L-GAL WITH SAMPLES FROM $\mathcal{G} = (\mathcal{V}, \mathcal{E})$

Often when trying to solve the L-GAL problem, one will be forced to work on sub-graph samples of the graph instead of the entire graph. This can happen for multiple practical reasons : (1) True graph is not known and we have access to the graph only through instances of sub-graphs. For example, true NLP graph is not known. But we have access to NLP text which hints at the graph. (2) Performing gradient descent on the entire graph is computationally prohibitive. For example social networking graphs are massive. In this case one need to sample nodes from $\mathcal{V}$ and then sample the neighborhood of these nodes to estimate $f_{agg}(\mathcal{N}(u))$ for all sampled nodes $u$. To account for this practical scenario, we propose an optimization objective for working with sub-graphs. Let $X$ be the data available for a particular domain, $\mathbb{G}_x = \mathcal{V}_x, \mathcal{E}_x$ be the sub-graph corresponding to the example $x \in X$. Let the neighborhood function restricted to this sub-graph be $\mathcal{N}_x$. We define the

optimization objective as,

$$\mathbf{E} = \arg\min_E \sum_{x \in X} \sum_{v \in \mathcal{V}_x} -\ln \mathcal{S}(E(v), f_{agg}(E(\hat{\mathcal{N}}_x(v)))) \tag{9}$$

where $\hat{N}(u)$ is neighborhood induced by the sub-graph. Essentially, we consider each node in the sub-graph and its induced neighbours $\hat{\mathcal{N}}$ from the sub-graph as an example and optimize the loss.

### 5.4 CSC WITH CONSTRAINTS

The problem of CSC , as are most other learning algorithms, is under-specified (particularly when $f_{agg}$ is a highly expressive function). Also, sometimes there is additional information that one wants to induct in the loss function which is not present in graph structure. For example, in commercial product search settings, "nike" and "adidas" get similar embeddings due to similar neighborhood graph structure but one might want that the embeddings learned distinguish between these entities. In such cases, a constraint needs to be imposed in CSC to minimize similarity between such pairs. This concept is generalized by using a negative sampling function NS $: \mathcal{V} \to 2^{\mathcal{V}}$ which defines a set of nodes in $2^{\mathcal{V}}$ on which a constraint needs to be imposed for any given node in $\mathcal{V}$. We write the problem of constrained CSC as follows:

$$\mathcal{C}^* = \arg\min_{\mathcal{C}} \sum_{u \in \mathcal{V}, v \in \mathrm{NS}(u)} -\ln(1 - \mathcal{S}(\mathcal{C}(u), \mathcal{C}(v))) \quad \text{subject to the CSC condition:}$$
$$\mathcal{S}(\mathcal{C}(u), \mathcal{C}(v)) = \mathcal{S}_{\mathcal{N}}(\mathcal{C}(\mathcal{N}(u)), \mathcal{C}(\mathcal{N}(v))) \,\forall u, v \in \mathcal{V}. \tag{10}$$

Using the Lagrangian multiplier we can re-write the constrained optimization objective of L-GAL as

$$\mathbf{E} = \arg\min_E \lambda \sum_{v \in \mathcal{V}} -\ln \mathcal{S}(E(v), f_{agg}(E(\mathcal{N}(v)))) + \sum_{v \in \mathcal{V}, u \in \mathrm{NS}(v)} -\ln(1 - \mathcal{S}(E(u), E(v))). \tag{11}$$

In most applications, $\lambda$ is set to 1. There are variants of inducing negative loss into a system, e.g., $\mathrm{softmax}$ (or sampled $\mathrm{softmax}$) is one such popular variant.

### 5.5 OPTIMIZATION ALGORITHM TO SOLVE L-GAL AND ITS VARIANTS

We have developed a series of optimization objectives to solve the problem posed by CSC in the previous subsections. We close this section by discussing the algorithms used to solve these optimization objectives. We can use any of the standard optimization algorithms: first order algorithms like gradient descent (Courant et al., 1994), stochastic gradient descent (Kiefer & Wolfowitz, 1952) or second order algorithms like Adam (Kingma & Ba, 2015), Adagrad (Duchi et al., 2011), etc. When solving problem on complete graph, we denote the algorithm as $\mathcal{A}(\mathcal{G} = (\mathcal{V}, \mathcal{E}), \mathcal{S}, f_{agg}, \mathrm{NS})$ which is parameterized by underlying graph $\mathcal{G} = (\mathcal{V}, \mathcal{E})$ , similarity metric $\mathcal{S}$, aggregation function $f_{agg}$, and negative sampling function NS $: \mathcal{V} \to 2^{\mathcal{V}}$. Whenever working with sub-graph samples of the true graph $\mathcal{G}$, we denote the algorithm as as $\mathcal{A}_s(X, \mathcal{S}, f_{agg}, \mathrm{NS})$ where $X$ is the data, each element $x$ of which, induces a sub-graph $\mathcal{G}_x$ ( The precise definition of this in provided in section 6.1)

## 6 INSTANTIATIONS OF L-GAL IN THE LITERATURE

Firstly, we describe the construction of the graph for a particular domain and sub-graph induction based on samples from the data. Secondly we summarize the graph construction and instantiations of L-GAL optimization objective for different embedding models in table 1 and discuss some aspects of it in section 6.2.

### 6.1 GRAPH CONSTRUCTION.

We provide a generic recipe of graph construction on structured domains. Consider for example a domain with token set $\mathcal{T}$. Our graph will have these tokens as nodes. However, we can introduce even higher order tokens by combining tokens. For example, the set $\mathcal{T}^2$ will denote a set of all bi-gram tokens in the domain. We can extend this idea to $n$-grams by considering the set $\mathcal{T}^n$. Thus, the set of all nodes in the graph of domain can be written as

$$\mathcal{V} = \cup_{i=1}^k \mathcal{T}^i, \tag{12}$$

Table 1: Examples of instantiation of $A_s$ for L-GAL for various state-of-the art embedding models - word2vec (Mikolov et al., 2013a), AWE (Sonkar et al., 2020), BERT (Devlin et al., 2019), ViT (Dosovitskiy et al., 2020), and Node2vec (Grover & Leskovec, 2016). Note that the token in BERT/ViT includes the position.

| Parameters of $\mathcal{A}_s$ | Word2Vec | AWE | BERT | ViT | Node2Vec |
|---|---|---|---|---|---|
| $\mathcal{T}$ : Tokens | words | words | word-pieces $\times \mathbf{N}$ | (16 x 16 patches) $\times \mathbf{N}$ | nodes |
| Gram-depth | 1 | 1 | BERT depth | ViT depth | 1 |
| $\mathcal{V}$ : Nodes | $\mathcal{T}$ | $\mathcal{T}$ | $\cup_{i=1}^{depth}\mathcal{T}^i$ | $\cup_{i=1}^{depth}\mathcal{T}^i$ | $\mathcal{T}$ |
| $\mathcal{E}$ : Edges | co-occurence with freq as weights | co-occurence with freq as weights | co-occurence with freq as weights | co-occurence with freq as weights | co-occurence in random walks with freq as weights |
| $\mathcal{S}(x,y)$ | $\exp\left(\langle \boldsymbol{x}, \boldsymbol{y}\rangle\right)$ | $\frac{1}{1+\exp\left(-\langle x,y\rangle\right)}$ | $\exp\left(\langle \boldsymbol{x}, \boldsymbol{y}\rangle\right)$ | $\exp\left(\langle \boldsymbol{x}, \boldsymbol{y}\rangle\right)$ | $\exp\left(\langle \boldsymbol{x}, \boldsymbol{y}\rangle\right)$ |
| Final loss with Negative Sampling loss | hierarchical softmax | sigmoid sampled loss | softmax | softmax | sampled softmax |
| Negative Sampling $\mathbf{NS}(u)$ | $\mathcal{T}/\{u\}$ | frequency based sampling to choose $\mathbf{NS}$ | $\mathcal{T}/\{u\}$ | $\mathcal{T}/\{u\}$ | random sample from $\mathcal{T}/\{u\}$ |
| $f_{agg}(\mathcal{N}(u))$ | $\sum_{v\in\mathcal{N}(u)} w_{(u,v)}\mathbf{E}(v)$ | $\sum_{v\in\mathcal{N}(u)} w_{(u,v)}\mathbf{E}(v)$ | $\sum_{v\in\mathcal{N}(u)} w_{(u,v)}\mathbf{E}(v)$ | $\sum_{v\in\mathcal{N}(u)} w_{(u,v)}\mathbf{E}(v)$ | $\sum_{v\in\mathcal{N}(u)} w_{(u,v)}\mathbf{E}(v)$ |
| $\hat{f}_{agg}(\mathcal{N}(u))$ | $\sum_{v\in\hat{\mathcal{N}}(u)} \mathbf{E}(v)$ | $\sum_{v\in\hat{\mathcal{N}}(u)} \hat{w}(u,v)\mathbf{E}(v)$ | $\sum_{v\in\hat{\mathcal{N}}(u)} \hat{w}(u,v)\mathbf{E}(v)$ | $\sum_{v\in\hat{\mathcal{N}}(u)} \hat{w}(u,v)\mathbf{E}(v)$ | $\sum_{v\in\hat{\mathcal{N}}(u)} \mathbf{E}(v)$ |
| $x \in \mathbf{X}$ x = subgraph induced on G=(V, E) by | words in a sentence | words in a sentence | word-pieces with position in a sentence | patches with position in an image | nodes in a random walk sequence |

where $k$ is the gram-depth. The edges in this graph of gram-depth $k$ depend on the application. This is illustrated in figure 1a. In most cases, including the models discussed in this paper, the edges are added on basis of co-occurrence statistics in the data. For example, let us consider the case of NLP. The sentence "The tree fell" in a graph with a gram-depth of 3 will include the nodes {the, tree, fell, {the, tree}, {the, fell}, {tree, fell}, {the, tree, fell}}. If the edges are added on the basis of co-occurrence, then based on the above sentence we add the edges between "The" and "fell," "The tree" and "fell", "The fell" and "tree," and so on. Additionally, edges can have weights corresponding to the frequency of co-occurrence.

While the graph on tokens represents the true graph, the algorithm $A_s$ which solves the sampled version of L-GAL problem uses the sub-graph of $\mathcal{G} = (\mathcal{V}, \mathcal{E})$ which is induced by the sample. Considering the sample $x \in X$ as a set of tokens $x \subseteq \mathcal{T}$, the graph induced by this set of tokens, $\mathcal{G}_x = (\mathcal{V}_x, \mathcal{E}_x)$ with neighborhood function $\mathcal{N}_x$ where

$$\mathcal{V}_x = \{v|v \in \cup_{i=1}^k x^i, v \in \mathcal{V}\}, \quad \mathcal{E}_x = \{(u,v)|u, v \in \cup_{i=1}^k x^i, (u,v) \in \mathcal{E}\}. \tag{13}$$

## 6.2 CURRENT STATE-OF-THE-ART EMBEDDING MODELS ARE INSTANTIATIONS OF L-GAL

Now, we summarize the instantiations of $\mathcal{A}_s$ for variety of algorithms in domains of NLP, images, and graphs. For details on working of each algorithm, we direct the reader to the corresponding original papers. We want to point out the usage of $\exp\langle \boldsymbol{x}, \boldsymbol{y}\rangle$ ( or $\frac{1}{1+\exp\left(-\langle \boldsymbol{x}, \boldsymbol{y}\rangle\right)}$ ) as similarity metrics. Under the unit-norm assumption on $\boldsymbol{x}$ and $\boldsymbol{y}$, a standard assumption for analysis of embeddings, $\exp\langle \boldsymbol{x}, \boldsymbol{y}\rangle = \exp\left(1 - \mathcal{D}^2(\boldsymbol{x}, \boldsymbol{y})/2\right)$ which is only a function of the $l_2$ norm $\mathcal{D}$ and inversely proportional to it. Thus, $\exp\langle \boldsymbol{x}, \boldsymbol{y}\rangle$ ( or $\frac{1}{1+\exp\left(-\langle \boldsymbol{x}, \boldsymbol{y}\rangle\right)}$ ) is a valid similarity metric under unit-norm assumption.

### 6.2.1 WORD2VEC, NODE2VEC, AWE

**Theorem 6.1.** The algorithm $\mathcal{A}_s$ for L-GAL initialized with parameters from table 1 for Word2Vec, Node2Vec and AWE leads to exactly the algorithms (with possibly minor variations) as proposed in the original papers of Word2Vec, Node2Vec and AWE

The proof of the above theorem is quite straightforward and is provided in the appendix. The formulation of sampled L-GAL problem provides a natural explanation of why AWE performs better than Word2Vec. Both Word2Vec and AWE operate on the same graph $\mathcal{G} = (\mathcal{V}, \mathcal{E})$ . However, they compute their estimates of $f_{agg}(\hat{\mathcal{N}}(u))$ differently. While word2vec performs simple sum, AWE performs weighted sum. One can prove that the MSE error with simple sum is larger than with weighted sum when weights are equal to the co-occurrence frequency. AWE models these weights as these weights are not known apriori and the authors in the paper hint at the learned weights being representative of the these co-occurrence frequencies.

Construction of Higher Order Graphs

(a) Graph Construction : higher order tokens in a graph

(b) Illustration of how Bert/ ViT follows L-GAL framework

### 6.2.2 BERT AND VIT

**Theorem 6.2.** Considering BERT and ViT as stacks of attention layers without any non-linearity and initializing the algorithm $A_s$ with the parameters mentioned in the table 1 leads to the algorithms (with possibly minor variations) as proposed in the original papers of BERT and ViT.

The proof of the theorem is provided in the appendix. For this analysis, we will assume that BERT is a stack of attention layers without any non-linearity. BERT and ViT have similar architecture and hence we provide a single proof for them. Let us consider the sub-graph induced for BERT by a sample $x$, which means it essentially has the nodes $\cup_{i=1}^k x^i$. The sub-graph will have every node connect with every other node. Thus when we look at the embedding of a particular masked word, according to definition of $f_{agg}$ for BERT we will get,

$$f_{agg}(\mathcal{N}(u)) = \sum_{v \in \mathcal{N}(u)} (w_{(u,v)} \mathbf{E}(v)), \qquad \hat{f}_{agg}(\mathcal{N}(u)) = \sum_{v \in \mathcal{N}(u) \cap (\cup_{i=1}^k x^i)} (\alpha_{(u,v)} \mathbf{E}(v)). \tag{14}$$

Now if this is exactly the computation that BERT (or ViT) performs then with softmax style negative sampling loss, BERT will follow the L-GAL optimization. We claim that BERT does indeed compute this $\hat{f}_{agg}$. Ideally, BERT should learn the embedding matrix for all the nodes including tokens and higher order tokens. However, this is computationally expensive. Hence, BERT actually models the higher order tokens in terms of its component tokens. It can be seen that if the modelling is a weighted linear sum of the components, then BERT essentially ends up computing equation 14. More details can be found in appendix. Also, as in AWE it models the weights in the summation of $f_{agg}$ (via the attention mechanism). The information propagated through BERT can be visualized as shown in figure 1b. At each layer $i \in \{1, .., k\}$, BERT computes the token embeddings of order $i$ and this information from each layer flows to the masked word node and gets aggregated.

## 7 IMPLICATIONS AND CONCLUSIONS

In this paper, we define a notion of CSC to understand the internals of wide range of embedding models across diverse structured domains. Grounded on this notion, we propose a generalized L-GAL optimization problem that solves for CSC on graphs induced by any structured domain. We thereby prove equivalence between loss functions of popular NLP, image, and graph embedding models and our proposed constrained L-GAL optimization loss operating on domain-specific graphs. The methodology of construction of these graphs is also presented for each structured domain.

Our proposed framework with its robust mathematical founding on CSC us graph theoretical perspective to these unsupervised embedding models. We have already seen great improvements in supervised models based on the connection between 1-WL and GNNs. We believe that our formulation will stimulate further research in embedding models. Insights from the underlying graph view of the computation in embedding models can give us new directions to improve quality and scalability and training efficiency of these models.

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

# A APPENDIX

## A.1 A WALK THROUGH POPULAR EMBEDDING MODELS AND UNDERSTANDING THEIR DESIGN PRINCIPLES AGAINST THE BACKDROP OF OUR FRAMEWORK

The proposed L-GAL formulation abstracts out different components of embedding models and thus enables design of better embedding models by focusing on these components. The components are as follows,

- Domain Graphs
- $f_{agg}$ Neighborhood aggregation function.
- $NS$ : Negative Sampling set.
- Estimation of $f_{agg}(\mathcal{N}(u))$ via sampling.

One can think of the strategies focusing on these components as ,

1. Better sampling/ estimates of a node's neighborhoods.
2. Better design of $f_{agg}$ function.
3. Construction of more expressive graphs that embed higher order tokens.
4. Importance based negative sampling to inject knowledge that best negates the neighbor information given by our proposed domain graph.

With these four strategies in place, next we understand the development of NLP models over time by choosing word2vec, one of the most widely-used embedding model, as our starting reference. We demonstrate how our proposed strategies have manifested in different embedding models that have improved over word2vec in last decade.

Recall from table 1, that word2vec uses

- $G = (V, E)$ where V is the set of all single words as nodes and E is the edge between words, say u and v, with co-occurrence as the weight.
- $f_{agg}(\mathcal{N}(u))$ weighted sum of neighbour embeddings
- NS(u): All words except u
- Sampling technique : sentences

**Strategy 1 : Better estimation of $f_{agg}(\mathcal{N}(u))$ - Glove vs Word2vec** Word2vec is an iterative algorithm over sentences. It uses sentence as a sample to guess word's neighborhood which induces a lot of noise. Our framework would suggest to use the entire computation of $f_{agg}$ instead of sampling if possible for a given $f_{agg}$ and graph. Glove does exactly this. Glove (Pennington et al., 2014a) improves upon word2vec by using global co-occurrence statistics to eliminate noise in these neighborhood estimates. Glove's optimization objective (given by equation 12 which is simplified to equation 15 in original paper) is as follows:

$$J = \sum_{i=1}^{V} \sum_{j=1}^{V} X_{ij} \log Q_{ij},$$

where $V$ is the vocabulary, $X_{ij}$ is the co-occurrence count of word $i$ with word $j$ in the corpus, and $Q_{ij} = \frac{\exp w_i^T w_j}{\sum_{k=1}^{V} \exp w_i^T w_k}$. Note that Glove simplifies $Q_{ij} = \exp w_i^T w_j$ which is identical to the similarity kernel we use in our analysis with unit norm assumption.

Strategy 1, thus, helps us to understand why Glove improves upon word2vec.

**Strategy 2 : Better $f_{agg}$ - AWE vs Word2vec** Let us now consider another word embedding model, AWE (Attention Word Embedding), that improves upon the CBOW model of word2vec. CBOW model weighs each word equally to compute the context embedding, however, AWE weighs

each context word differently as some context words are more important for prediction of the masked word than others. The context vector in AWE is given by:

$$c_{\mathbf{awe}} = \sum_{i \in [-b,b]-\{0\}} a_{w_i} u_{w_i} \text{ where } a_{w_i} = \exp\left(k_{w_0}^T q_{w_i}\right) ,$$

where where $b$ is the size of the context window and $w_i$ is the index of each word ($w_0$ is the index of the masked word; the rest are the indices of the context words), $a_{w_i}$ is the attention weight of each context word vector $u_{w_i}$ calculated using the key matrix $K$ and the query matrix $Q$. Note that in word2vec $a_{w_i} = 1$.

Strategy 2 helps us understand why AWE outperforms Word2vec. AWE formulates a better $f_{agg}$ function as compared to Word2Vec.

**Strategy 3 : More informative domain graphs - BERT vs Word2vec**   As discussed in section **??**, BERT uses a graph with higher order tokens. In fact, the highest order of token can be interpreted as the number of layers in ther BERT. It also uses the AWE style weighting of the neighbours. Strategy 3 (along with strategy 2) can help us understand how BERT improves over Word2vec embedding model. Strategy 3 can also help us understand why (Le & Mikolov, 2014; Tang et al., 2015; Liu et al., 2015) can improve over Word2Vec.

**Strategy 4 : Better $NS$ - WSD-W2V vs Word2Vec**   Strategy 4 helps to understand yet another shortcoming of word2vec. Word2vec struggles with modeling antonyms and often brings antonyms close to each other (Thalenberg, 2016). This is not surprising since antonyms tend to occur in similar contexts, i.e., identical neighborhoods. With our proposed framework, one can arrive at this conclusion from a graph theoretic perspective. Antonym subgraphs are isomorphic, and thus antonyms get assigned similar continuous stable colors. To overcome this, one needs to introduce information into the setup using variants of negative sampling. Thus external datasets like wordnet (Orkphol & Yang, 2019) need to complement word2vec's training strategy for efficient modeling of antonyms.

To conclude, the model design changes that our framework suggest (strategies 1-4) can be found in existing works.

## A.2   THEOREM 4.1

Let $\{w_1, w_2, .., w_n\}$ be words in vocabulary, and $\mathbb{M} \in \mathbb{R}^{n \times n}$ is a co-occurrence matrix with entries $e_{ij} \in \mathbf{N}$ containing the number of times $w_i$ co-occurs with $w_j$ within a fixed context window. Let there be a function $f : w \to \mathbf{N}$ which takes a word and maps it to a color $c \in \mathbf{N}$ (assigns meaning in accordance with Distributional Hypothesis), such that $f(w_i) = f(w_j)$ only if row $i$ is same as row $j$ in matrix $\mathbb{M}$. Construct a graph $G_{DH}$ with words as nodes and its adjacency matrix given by $\mathbb{M}$. Then, function $f$ defines a stable coloring on graph $G_{DH}$.

**Proof:** The proof follows trivially from the definition of $f$.

## A.3   THEOREM 4.2

Consider a discrete signal $y(x)$ sampled at $n$ points $x_i$ ($i = 1, ..., n$), and let the sequence $p(x_i) = (y(x_{i-t}), .., y(x_{i-1}), y(x_{i+1}), .., y(x_{i+t}))$ be a patch of neighborhood values around each $x_i$ for some context window length $t$. NLM denoises the signal $y(x_i)$, $i = 1, ..., n$ with iterative updates. The fixed point denoised version of the signal $y_d$ can be written as follows:

$$y_d(x_i) = \frac{1}{D(x_i)} \sum_{j=1}^{n} K(\boldsymbol{p}_d(x_i), \boldsymbol{p}_d(x_j)) y_d(x_j), \tag{15}$$

where $D(x_i) = \sum_{j=1}^{n} K(y_d(x_i), y_d(x_j))$ and K is an arbitrary kernel function. Let the graph $\mathcal{G}_{nlm} = (\mathcal{V}_{nlm}, \mathcal{E}_{nlm})$ where each $x_i$ is a node $u_i$ in $\mathcal{V}_{nlm}$ and each pair $(u_i, u_j)$ with $|i - j| \leq t$ is represented as a directed edge in $\mathcal{E}_{nlm}$ with label $(i - j)$. Then the fixed point solution of NLM with Kronecker delta kernel $K_\delta$, $y_d : \mathbf{N} \to R$ defines a *weak*-stable coloring over the graph $\mathcal{G}_{nlm}$

**Proof:**

In order to prove a weak coloring, we need to prove that if the neighbourhoods of a pixel are equal then its value is equal. Consider any two pixels $x_i$ and $x_j$ with same neighbourhoods, Then $K_\delta(p_d(x_i), p_d(x_j)) = 1$. Let $\{x_{k_1}, x_{k_2}, x_{k_3}, ...\}$ be the set of size L of pixels whose neighbourhoods that match with neighbourhood of $x_i$ and $x_j$. Thus, by equation of final stable solution, both the pixel values are equal to $y_d(x_i) = y_d(x_j) = \frac{1}{L} \sum_{l=1}^{L} K_\delta(\boldsymbol{p}_d(x_i), \boldsymbol{p}_d(x_{k_l})) y_d(x_{k_l})$

### A.4  THEOREM 5.1

(SC is a special case of CSC ). Stable coloring (discrete) problem is an instance of continuous stable coloring problem with $\mathbf{L} = \mathbf{N}$, $\mathcal{S}(i, j) = \mathbb{1}(i = j)$ and $\mathcal{S}_\mathcal{N}(s_1, s_2) = \mathbb{1}(s_1 = s_2)$ where $i, j \in \mathbf{N}$ and $s_1, s_2 \in \mathbf{N}^\mathbf{N}$. In this case $f_{agg} : \mathbf{N}^\mathbf{N} \to \mathbf{N}$ function is essentially an injective hash function which maps multi-subsets of $\mathbf{N}$ to $\mathbf{N}$.

**Proof:** The proof of the theorem trivially follows by using the parameters of CSC as mentioned in the theorem.

### A.5  THEOREM 5.2

Let the solution $\mathbf{E}$ to L-GAL upper-bounds each term in the summation of loss in equation 8 by some $\epsilon > 0$; thus it upper-bounds total loss by $|\mathcal{V}|\epsilon$. Then the same solution matrix $\mathbf{E}$ is a solution to G-GAL with each term in summation upper-bounded by $2\epsilon$ and thus upper-bounding the total loss by $|\mathcal{V}|^2\epsilon$.

**Proof:**

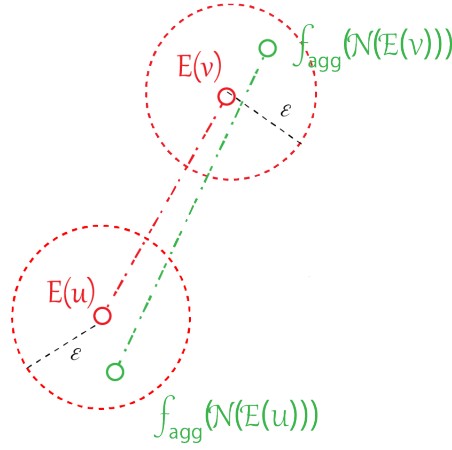

Figure 2: Illustration of theorem 5.2

$$Loss(\text{G-GAL }) = \sum_{u,v \in \mathcal{V}} abs(-\ln \mathcal{S}(E(u), E(v)) + \ln \mathcal{S}(f_{agg}(E(\mathcal{N}(u))), f_{agg}(E(\mathcal{N}(v))))) \tag{16}$$

$$= \sum_{u,v \in \mathcal{V}} abs(\mathcal{D}(E(u), E(v)) - \mathcal{D}(f_{agg}(E(\mathcal{N}(u))), f_{agg}(E(\mathcal{N}(v))))) \tag{17}$$

As $\mathcal{D}$ is a metric, we have

$$\mathcal{D}(f_{agg}(E(\mathcal{N}(u))), f_{agg}(E(\mathcal{N}(v)))) \le \mathcal{D}(f_{agg}(E(\mathcal{N}(u))), E(u)) + \mathcal{D}(E(u), E(v)) + \mathcal{D}(E(v), f_{agg}(E(\mathcal{N}(v)))) \tag{18}$$

$$\mathcal{D}(f_{agg}(E(\mathcal{N}(u))), f_{agg}(E(\mathcal{N}(v)))) - \mathcal{D}(E(u), E(v)) \le \mathcal{D}(f_{agg}(E(\mathcal{N}(u))), E(u)) + \mathcal{D}(E(v), f_{agg}(E(\mathcal{N}(v)))) \tag{19}$$

By hypothesis, that the each term in the loss of L-GAL is bound by $\epsilon$

$$\mathcal{D}(f_{agg}(E(\mathcal{N}(u))), f_{agg}(E(\mathcal{N}(v)))) - \mathcal{D}(E(u), E(v)) \leq 2\epsilon \tag{20}$$

Thus ,

$$Loss(\text{G-GAL }) \leq \sum_{u,v \in \mathcal{V}} 2\epsilon = |\mathcal{V}|(|\mathcal{V}| - 1)\epsilon \leq |\mathcal{V}|^2\epsilon \tag{21}$$

## A.6 Theorem 6.1

**Theorem A.1.** The algorithm $\mathcal{A}_s$ for L-GAL initialized with parameters from table 1 for Word2Vec, Node2Vec and AWE leads to exactly the algorithms (with possibly minor variations) as proposed in the original papers of Word2Vec, Node2Vec and AWE

**proof** It is easy to check that the parameters specified in the table 1 exactly provides the algorithms of corresponding models. Some more details can be found in appendix B

## A.7 Theorem 6.2

Considering BERT and ViT as stacks of attention layers without any non-linearity and initializing the algorithm $A_s$ with the parameters mentioned in the table 1 leads to the algorithms (with possibly minor variations) as proposed in the original papers of BERT and ViT with the linear modelling on embedding of higher order tokens.

**Proof**: The parameters of BERT / ViT such as loss used, tokens can be verified from the table. It only remains to prove that BERT computation is same as that of $f_{agg}$ mentioned.

This is done under two assumptions, (1) BERT is just a stack of attention layers. (2) each higher order token's embedding is modelled as a linear model of it constituent embeddings.

Let us consider the token $(u, i)$ to be masked. Without loss of generality let the position in the token be last position. Let each path reaching the level k at masked word be denoted with the locations of the words the path takes at each level. So a path of (0,1,0,2) says that it takes the route of words $w_0, w_1, w_0, w_2$ to reach the masked word at the final level.

Then we can write the new computation of reaching BERT at level k as

$$\sum_{p \in PATHS(k)} \sum_{i=1}^{k} \alpha_{ip} E(w[p(i)]) \tag{22}$$

This can be seen as the embedding of the path as BERT models it (linear model assumption). It is easy to see that each path of length k maps one-to-one to a higher order token in $\mathcal{T}^k$. Under this relation PATHS(k) is exactly the set of all nodes in $\mathcal{S}^k$ The information also flows from masked words which aggregates the embeddings of $\mathcal{S}^i, i < k$. Thus the final computation of bert at level K looks like

$$\sum_{k=1}^{K} \sum_{p \in PATHS(k)} \sum_{i=1}^{k} \alpha_{ip} E(w[p(i)]) = \sum_{v \in S^K} \beta_v E(v) \tag{23}$$

for some $\beta$s

## B Examples of graph construction

### B.1 Various domains as graphs

Any general set of tokens say $\mathcal{T}$ and associated relation $R \subseteq \mathcal{T} \times \mathcal{T}$ can give us the graph $\mathcal{G} = (\mathcal{V}, \mathcal{E})$ with $\mathcal{V} = \mathcal{T}$ and $(u, v) \in \mathcal{E}$ iff $R(u, v) = 1$. While the underlying graph of NLP is not explicitly known some examples of relations that can be drawn from the sentence structure are as follows.

- Let $\mathcal{V}$ be the set of all words $\{w_1, w_2, ...w_{|\mathcal{V}|}\}$. $(w_1, w_2) \in \mathcal{E}$ iff the words $w_1$ and $w_2$ co-occur in a sentence. Additionally, we can assign a weight to the edge equal to the frequency of co-occurrence observed in some natural language corpus.
- Let $\mathcal{V}$ be a set of all subsets of $\mathcal{T}$ of size less than or equal to $k$. Let $t_1, t_2 \in \mathcal{V}$ then the relation $(t_1, t_2) \in \mathcal{E}$ iff $t_1 \cup t_2$ appear together in some sentence in natural language.

Similarly we can also view the entire image collection as a graph $\mathcal{G} = (\mathcal{V}, \mathcal{E})$ where $\mathcal{V}$ is a set of all $k \times k$ patches observed in all images ( or alternatively images of a particular category ). Some examples of relations might be,

- $R(p_1, p_2) = 1$ iff $p_1$ occurs in a larger patch $K \times K$ around $p_2$ in some images
- $R(p_1, p_2) = 1$ iff $p_1$ adjacent to $p_2$ in four directions (left, right, top, down)

Graph domain naturally maps to the graph data structure. However, we can construct more informative graphs from the basic graph structure. For example, we can add hyper-edges to add hyper-graph structure to the same graph.

## B.2  WORD2VEC

Word2vec is one of the standard word embedding models (Mikolov et al., 2013a;b). The continuous bag-of-words (CBOW) model of word2vec predicts a masked word using its context. Let $\mathbf{E} = [\boldsymbol{v}_1, ..., \boldsymbol{v}_N]^T \in \mathbb{R}^{N \times d}$ be the word embedding matrix where $N$ is the size of the vocabulary, and $d$ is the size of the word vector. CBOW looks at the set of sentences drawn from NLP corpus $\mathbf{X}$ and uses it to learn $\mathbf{E}$. For each sentence $x$ of size $n$, CBOW creates $n$ examples of $(u, \text{ctx}(u) = \{v|v \in x, v \neq u\})$ pairs by choosing one word and using rest of the words as context ctx.

It uses the following formulation,

$$\mathbf{E} = \underset{E}{\arg\min} \sum_{(u,\text{ctx}(u) \in x \in \mathbf{X})} \left( -\ln \frac{1}{1 + \exp\left(-\langle E(u), E(\text{ctx}(u)) \rangle\right)} + \tag{24}$$

$$\sum_{v \in \text{NS}(u)} -\ln\left(1 - \frac{1}{1 + \exp\left(-\langle E(v), E(\text{ctx}(u)) \rangle\right)}\right)\right) \tag{25}$$

where $E(\text{ctx}(u)) = \sum_{v \in \text{ctx}(u)} E(v)$. Note that original formulation of CBOW uses softmax which is a variant of the negative sampling loss formulation.

We will show in theorem **??** that Word2Vec is indeed same as $\mathcal{A}_s$ using some specific initialization of its parameters. We first define the parameters.

- **Graph** $\mathcal{G}_{w2v}$. Let $\{w_1, w_2, .., w_{|V|}\}$ be words in vocabulary $V$, and $\mathbb{M} \in \mathbb{R}^{|V| \times |V|}$ is a co-occurrence matrix with entries $e_{ij} \in \mathcal{N}$ containing the number of times $w_i$ co-occurs with $w_j$ within a fixed context window. Construct a graph $G_{w2v}$ with words as nodes and its adjacency matrix given by $\mathbb{M}$
- **Data** $\mathbf{X}_{w2v}$. Every example is a set of words (sentence) and the graph induced by these words is a sub-graph in $\mathcal{G}_{w2v}$ with every node connecting every other node.
- **Aggregation function** $f_{w2v}$. Let $f_{agg}(\text{ctx}(u)) = \sum_{v \in \text{ctx}(u)} E(v)$.
- **Similarity Metric** $\mathcal{S}_{sgm}$. Let $\mathcal{S}(x, y)$ be $\sigma(x, y)$. Under unit-norm assumption, this is a valid similarity metric derived from $l_2$ norm distance metric.

## B.3  ATTENTION WORD EMBEDDINGS (AWE)

As can be observed in $f_{w2v}$, CBOW model equally weights the context words when making a prediction. To address this limitation of CBOW, AWE augments CBOW model with an attention mechanism to attend to context words that are most relevant for prediction of masked word. AWE also uses the same formulation as word2vec given by equation 25, except that context vector is a weighted sum of context word embeddings.

In AWE, $E(\text{ctx}(u)) = \sum_{v \in \text{ctx}(u)} w_{uv} E(v)$, where $w_{uv} = \exp\left(< \boldsymbol{k}_u, \boldsymbol{q}_v >\right)$ models the importance of context word $v$ for predicting the masked word $u$ using key and query word embedding matrices given by $\mathbf{K} = [\boldsymbol{k}_1, ..., \boldsymbol{k}_N]^T \in \mathbb{R}^{N \times d}$ and $\mathbf{Q} = [\boldsymbol{q}_1, ..., \boldsymbol{q}_N]^T \in \mathbb{R}^{N \times d}$ respectively.

As we did for word2vec, we will show in theorem **??** that AWE is indeed same as $\mathcal{A}_s$ using some specific initialization of its parameters. We first define the parameters.

- **Graph** $\mathcal{G}_{awe}$. $G_{awe} = G_{w2v}$.
- **Data** $\mathbf{X}_{awe}$. Similar to word2vec, every example is a set of words (sentence) but unlike the case of word2vec, the graph induced by these words is a **weighted** sub-graph in $\mathcal{G}_{awe}$.
- **Aggregation function** $f_{awe}$. Let $f_{agg}(\text{ctx}(u)) = \sum_{v \in \text{ctx}(u)} w_{uv} E(v)$.
- **Similarity Metric** $\mathcal{S}_{sgm}$. Let $\mathcal{S}(x, y)$ be $\sigma(x, y)$. Under unit-norm assumption, this is a valid similarity metric derived from $l_2$ norm distance metric.

