# OpenReview forum: "Embedding models through the lens of Stable Coloring"
_ICLR.cc/2022/Conference — ICLR 2022 Submitted_

### Official Review · Reviewer_K2fA · 2021-10-27

**Correctness:** 3
**Technical Novelty And Significance:** 2
**Empirical Novelty And Significance:** 1
**Recommendation:** 3
**Confidence:** 4

**Main Review:**

Major comments
--------------
- One of the main issues with this paper is that writing is not clear. In fact, the presentation of the paper leaves the reader with a feeling that this was an initial draft of the work. The paper is pretty difficult to read. There are many typos and the structure is disorganized. I suggest the authors work on improving it.

- Overall, the paper seems to be proposing an interesting contribution. The authors establish a connection between the Weisfeiler-Lehman test of isomorphism and embedding algorithms from structured domains. However, in my view the significance of the paper is low. The authors provide no discussion on potential applications of the proposed methodology, while it is not clear whether it can be useful for domains in which there is already a wealth of embedding approaches such as in natural language processing or in graph representation learning. Furthermore, the authors do not experimentally evaluate the proposed methodology.

- It is not clear to me what is the intuition behind the G-GAL and L-GAL formulations, and how exactly the solutions of those formulations are equivalent to the continuous stable coloring. I would suggest the authors make clear why out of all possible formuations, they have proposed those two and not some other formulation. Furthermore, if we assume that $f_{agg}$ is the sum operator, to my understanding, these two problems admit a trivial solution. Setting the representations of all nodes equal to the all-zeros vector leads Equations (7) and (8) to their minimum value (i.e., a value equal to 0). I would like the authors to comment on this point.

- It is not clear to me how embedding algorithms such as word2vec and node2vec are instances of the L-GAL formulation. In the case of those algorithms, a positive instance cooresponds to a pair of words/nodes that co-occur within a fixed-size window in some text or a random walk. In Equation (8), the representation of a node is encouraged to be close not to the representation of each of its neighbors, but to the sum of them (in case $f_{agg}$ denotes the sum operator).

- The authors present in section 5.5 the different optimization algorithms that one could employ to solve the L-GAL problem and its variants. It would strengthen a lot the paper if the authors could apply the proposed methodology to some real-world problem. For instance, the authors could construct a graph from some text corpus and use the proposed approach to embed the different term to some vector space. Then, they could also evaluate the quality of the generated embeddings in standard tasks such as word analogy and word similarity, and compare them againt those generated by other methods.

- Although I am not aware of any previous studies that have established a connection similar to the one presented in this paper, there are previous studies that have proposed to unify embedding approaches into matrix factorization frameworks [1]. Such works are related to the work presented in this paper and should be discussed in the related work section.

Minor comments
--------------
- Function $\mathcal{C}$ which is used for comparing vectors or multisets of vectors seems quite arbitrary. What are other functions that could potentially be employed and what properties do they need to exhibit?

- In p.2, the following sentence "As mentioned ... information?" is very long. It is also not clear and does not read well. I suggest the authors rephrase this sentence.

- In p.3, the authors mention "Chen et al. (2019) in their GIN ...". In fact, GIN was proposed by Xu et al. and not by Chen et al.

Typos:
------
p.2: "architectures captures capture the" -> "architectures capture the"\
p.3: "$\mathcal{C}_( \mathcal{N}(u)$" -> "$\mathcal{C}_i (\mathcal{N}(v))$"

p.3: "examples are given in appendix ??" -> "examples are given in appendix B"\
p.4: "surrounding it" -> "surrounding them"\
p.4: "provided in appendix" -> "provided in appendix."\
p.5: "which we show this in the" -> "which we show in the"\
p.5: "injective $f_{merge}$" -> "injective $f_{update}$"\
p.6: "learn-able" -> "learnable"\
p.8: "heirarchial" -> "hierarchical"

[1] Qiu, J., Dong, Y., Ma, H., Li, J., Wang, K. and Tang, J., Network embedding as matrix factorization: Unifying deepwalk, line, pte, and node2vec. In Proceedings of the 11th ACM International Conference on Web Search and Data Mining, pp. 459-467, 2018.

**Summary Of The Paper:**

In this paper, the authors investigate how the solution of the Weisfeiler-Lehman test of isomorphism (WL) is related to embedding algorithms from structured domains. The authors generalize the WL algorithm to the case where the colors assigned to the nodes correspond to feature vectors, and propose a series of optimization problems to solve this problem. Finally, they show that widely used algorithms in natural language processing, computer vision and graph representation learning such as word2vec, BERT, Visual Transformer and node2vec actually solve different instantiations of this common optimization problem.

**Summary Of The Review:**

In general, I feel that there is some value in this paper and that the contribution is interesting since,  to my knowledge, no paper so far has established any connection between the Weisfeiler-Lehman test and embedding approaches. However, I have several concerns about this work. The writing is not clear, while the significance of the theoretical results seems to be low. The authors do not perform any experiments, thus there are no empirical results and it is not clear how the proposed approach would work in real-world scenarios. Finally, I am not sure whether some of the claims made by the authors are actually true.

---

> ### Author Response · Authors · 2021-11-19
> **Clarifications and additional discussion of evidence in support of theory**
>
> Thank you for your comments. There was a general concern among reviewers that there is a lack of experimental evidence to support the theory or its implications. We do not perform experiments because we believe that there is ample evidence in existing literature that supports the implications of our theory. **We have added a new section in appendix (A.1 in blue)**. In this new section, we do a walkthrough of NLP embedding models and discuss how various improvements in NLP embedding models can be explained via implications of CSC based framework.
>
>
> **Contribution and Utility :**
> We unify the different conjectures such as - distributional hypothesis, non-local-means , etc (which are the basis of different algorithms in literature) into a single mathematical problem of continuous stable-coloring. This problem leads to formulation of Global GAL framework. We show that current state-of-the-art algorithms in these fields solve different instantiations of  Local GAL problem which gives a solution that is provably close to the solution of Global GAL problem which. This generic framework highlights that art of embedding model design lies in devising good underlying domain graphs and $f_{agg}$ functions.
>
> Against this backdrop, we can evaluate the existing algorithms and understand them better. See newly added section A.1 in appendix. For example, “Why does BERT perform better than Word2Vec?” -  Using different layers, BERT can be interpreted to be using higher order tokens in the underlying graph. While word2vec uses 1-gram tokens, BERT with $L$ layers uses upto $L$-gram tokens. Thus, one can expect BERT to perform better as it incorporates more information.
>
> We can also use this framework to seek further research directions. For example,
> “How can we improve an algorithm like word2vec ?“ - Word2vec improvements can come from a variety of considerations such as
> Can we have a better $f_{agg}$ ? One can see AWE as implementing better $f_{agg}$ over word2vec and we see that AWE beats word2vec
> Can we have a better estimation of chosen $f_{agg}$. Right now we only consider sentences of fixed context. This only gives us an estimation of $f_{agg}(N(u))$ as it samples from the neighbourhood. However, can we have better sampling?
> Can we better model the underlying graph? One can see that BERT considers more informative graph and beats Word2vec
>
> **It is not clear to me what is the intuition behind the G-GAL and L-GAL formulations.**
> Global-GAL formulation is exactly the solution to the CSC problem. The term global GAL minimizes is exactly $abs(ln( (S(u,v) / s_N(N(u),N(v)))$  i.e. it tries to make S(u,v) = S_N(N(u), N(v)).
> Local-GAL formulation decouples the problem while maintaining theoretically provable approximation to the Global-GAL solution.
> Trivial solution
> Trivial solution will not happen when the $f_{agg}$ is injective.. If $f_{agg}$ is not injective (e.g., mean, sum, median), then trivial solutions will exist for the optimization problem.
>
> **It is not clear to me how embedding algorithms such as word2vec and node2vec...**
> We consider the CBOW model for word2vec and node2vec. We have clarified this now. Skip-gram models can also be explained in this framework.

---

> > ### Comment · Reviewer_K2fA · 2021-11-22
> > **Re**
> >
> > I would like to thank the authors for their response. However, most of my concerns were not properly addressed. Therefore, I will keep my current rating.

---

### Official Review · Reviewer_mH7f · 2021-10-27

**Correctness:** 3
**Technical Novelty And Significance:** 2
**Empirical Novelty And Significance:** Not applicable
**Recommendation:** 3
**Confidence:** 3

**Main Review:**

Strengths:

1) The main contribution of the paper is the definition of a new problem CSC. As the authors themselves mention most of their Theorems are straightforward consequences of the definition of the problem.

Major Weaknesses

1) **No new insights**

The major weakness of the paper, is despite making a connection between CSC and prior word embedding techniques, the paper does not mention anything new about the techniques or present any insights into how the new connections can be used to better design embedding techniques.

Specifically, the main connection here is that we embed words by tryin to predict the context of the word and that this is the same as embedding a graph by trying to predict the neighbors of the graph. However, this connection (while not presented as CSC) has been known. See [1] for an example that explores the connection to 1WL in detail.

Further, my main issue with the paper is that the connection between techniques and word2vec has been made at a surface level in function space. That is, it shows that the embeddings that the two techniques are trying to learn are similar. However, it does not prove any properties  about the embeddings. Maybe say something about its generalization. Or show that if the graph satisfies certain properties then the learned embedding satisfies certain properties.

Further, the more interesting question in deep learning and the usage of neural networks is not the connection in function space, but the connection in parameter space. That is, how do these networks parameterize the function that we are trying to learn/how do optimization techniques learn different representations in the parameter space.

[1] Martin Grohe. 2020. Word2vec, node2vec, graph2vec, X2vec: Towards a Theory of Vector Embeddings of Structured Data. In Proceedings of the 39th ACM SIGMOD-SIGACT-SIGAI Symposium on Principles of Database Systems (PODS'20). Association for Computing Machinery, New York, NY, USA, 1–16. DOI:https://doi.org/10.1145/3375395.3387641

2) **Missing Details**

I also think the paper is missing details. First, in the problem of CSC, we look at $f_{agg}$ as an *injective* function from the space of all multisubsets of $L$ to $L$. However, the space of *all* multisubsets of $L$ has a strictly bigger cardinality than $L$. (eg. if $L$ is the natural numbers, i.e., is countable, then $N^L$ is uncountable). Hence no injective function can exist. Hence I assume the authors mean that the $f_{agg}$ is defined on the space of all *finite* multisubsets. While this is not too big a deal, it is an issue for a theoretical paper.

There are also a few vague statement. For example, the authors, say $f_{agg}$ is essentially a hash function in Theorem 5.1. What does this mean exactly? Another is after the box titled research question, the authors say, "the above mentioned issue of multiplying noisy neighborhoods". Where are we multiplying neighborhood? What does it mean to multiply neighborhoods?


**Minor Details**

There are typos in the main text. Some of them are

1) Many equations are missing parenthesis. For example, we see $\mathcal{C}(\mathcal{N}(u)$ in a few places (def 2.1, top of page 3, last line of 1WL paragraph).
2) Equation 1, also seems to have a typo, it should be $E^{k-1}(v) : v \in \{\{\mathcal{N}(u)\}\}$.
3) Theorem 4.2 last line has a typo.
4) Just before section 3, missing a reference (have ??)

**Improvements**

Here are just a few ideas for improvements.

1) Maybe look into $f_{agg}$ and see how properties of this function effect properties of learned embedding.
2) Run some experiments. Show empirically, how optimizing for CSC directly using the algorithms presented has similar results to doing the appropriate word embedding technique.
3) Show if the graph has certain properties then we can say something about the embedding.
4) Show that using insights from above we can design a new technique.



**Summary Of The Paper:**

In the paper the authors define a new problem called the continuous stable coloring (CSC). This is an extension of the traditional stable coloring problem, however, instead of having discrete color labels, you have continuous color labels. The authors then provide an objective function that could be minimized to obtain a CSC. They then show that if you tweak the similarity measure $S$ and the aggregator function $f_{agg}$, then the CSC problems various word embedding techniques such as word2vec, AWE, Bert (without non-linearity).

**Summary Of The Review:**

Overall, I think the problem of CSC is a neat problem. However, its connection to word embedding techniques has only been explored at a surface level. Hence I feel there is not enough in the paper to merit acceptance as a theory paper. However, the connection could bear fruit if the authors do provide deeper connections either in parameter space or provide results on the properties of the embedding learned. While I cannot vote for acceptance at this time, I do hope the authors revise with new results and resubmit.

---

> ### Author Response · Authors · 2021-11-19
> **Clarifications and additional discussion of evidence in support of theory**
>
> Thank you for your comments. There was a general concern among reviewers that there is a lack of experimental evidence to support the theory or its implications. We do not perform experiments because we believe that there is ample evidence in existing literature that supports the implications of our theory. **We have added a new section in appendix (A.1 in blue)**. In this new section, we do a walkthrough of NLP embedding models and discuss how various improvements in NLP embedding models can be explained via implications of CSC based framework.
>
> **Contribution and Utility :**
>
> We unify the different conjectures such as - distributional hypothesis, non-local-means , etc (which are the basis of different algorithms in literature) into a single mathematical problem of continuous stable-coloring. This problem leads to the formulation of the Global GAL framework. We show that current state-of-the-art algorithms in these fields solve different instantiations of  Local GAL problem which gives a solution that is provably close to the solution of Global GAL problem which. This generic framework highlights that art of embedding model design lies in devising good underlying domain graphs and $f_{agg}$ functions.
>
> Against this backdrop, we can evaluate the existing algorithms and understand them better.See newly added section A.1 in appendix For example, “Why does BERT perform better than Word2Vec?” -  Using different layers, BERT can be interpreted to be using higher order tokens in the underlying graph. While word2vec uses 1-gram tokens, BERT with L layers uses upto L-gram tokens. Thus, one can expect BERT to perform better as it incorporates more information.
>
> We can also use this framework to seek further research directions. For example,
> “How can we improve an algorithm like word2vec?” - Word2vec improvements can come from a variety of considerations such as
> Can we have a better $f_{agg}$ ? One can see AWE as implementing better $f_{agg}$ over word2vec and we see that AWE beats word2vec
> Can we have a better estimation of chosen $f_{agg}$. Right now we only consider sentences of fixed context. This only gives us an estimation of $f_{agg}(N(u))$ as it samples from the neighbourhood. However, can we have better sampling?
> Can we better model the underlying graph? One can see that BERT considers more informative graph and beats Word2vec
>
> **On 1-WL and L-GAL**
>
> The connection that the reviewer highlights between 1-WL and GNN layers is explored well in literature. However, it is only explored only in context of the question "What architecture should a particular network have w.r.t to two considerations - generalizability and expressivity".
>
> However, this is an entirely orthogonal discussion which, when looked at in context of our paper, tries to answer the question "what is a good $f_{agg}$". Contrary to this, in our paper, we connect the solutions of embedding models to stable-colored solutions in continuous domains.
>
> **On other embedding properties**
>
> While these are interesting questions, we believe that what we present is the starting point where we look at principles underlying state-of-the art methods. This, for us, is quite interesting. To the best of our knowledge, we do not know of a work that unifies the different conjectures such as - distributional hypothesis, non-local-means, etc (which are the basis of different algorithms in literature) into a single mathematical problem of continuous stable-coloring.
> We believe that our work can be a solid starting point to incorporate more properties of embeddings and analyse them through our framework.
>
> **Minor questions**
> 1) Uncountable number of multi-subsets : We deal with finite graphs and hence, we are looking at finite multi-subsets. We will clarify this in our statement.

---

### Official Review · Reviewer_EdGh · 2021-11-01

**Correctness:** 2
**Technical Novelty And Significance:** 2
**Empirical Novelty And Significance:** Not applicable
**Recommendation:** 3
**Confidence:** 3

**Main Review:**

- The writing of the paper can be improved. In particular, the contribution of this paper is unclear to me. The authors repetitively claim that they "define a mathematical framework that defines continuous stable coloring on graphs and develops optimization problems to search for them", but there is no validating algorithm at all.

- From a theoretical point of view, how could the proposed mathematical framework help us better understand NLP embedding models?

- It seems the authors try to claim that there exists some similarity between graph coloring problems and NLP embedding problems. However it feels like they are simply repeating the so-called distributional hypothesis. Now say the claim is true. How would that be helpful to researchers using those NLP models, including BERT, ViT, Node2Vec? Again the contribution is not clear.

- I do not understand if Theorem 4.1 is a theorem, or a hypothesis or an assumption. Is G_{DH} a weighted graph? Considering that the entries in M are nonnegative integers (counting co-occurrence), what does it mean by adjacency matrix in this case?

- Many Theorem proofs are just one line like "the proof follows trivially from ..." E.g., Theorem 4.1. If they are so obvious, is it necessary to present them as "Theorems"?

- The derivation from the assumptions in Theorem 4.1 to Definition 2.1 is not so trivial. How do you define neighborhoods in terms of the co-occurrence matrix?

- Section 2.2, description of 1-WL algorithm "Ci(N(u)) 6= C(N(u)". This is not properly typed. Why both sides are u? Also, which function C is being used on the RHS? Please clarify.

Editorial comments:

- Extra whitespace: introduction - "continuous stable coloring (CSC )", Section 2 - "Aggregate and Update (GAU )", Section 2.2 "(GAU )".

- Section A.6: "with the parameters mentioned in the table ??"

- Definition 2.1: "C(N(u)) = C(N(v)"

**Summary Of The Paper:**

The paper tries to relate multiple unsupervised NLP embedding models to the problem of stable coloring. The authors claim that they "prove equivalence between loss functions of popular NLP, image, and graph embedding models and our proposed constrained L-GAL optimization loss operating on domain-specific graphs".

**Summary Of The Review:**

At the current stage the contribution of this paper is unclear, both theoretically and experimentally.

---

> ### Author Response · Authors · 2021-11-19
> **Clarifications and additional discussion of evidence in support of theory**
>
> Thank you for your comments. There was a general concern among reviewers that there is a lack of experimental evidence to support the theory or its implications. We do not perform experiments because we believe that there is ample evidence in existing literature that supports the implications of our theory. **We have added a new section in appendix (A.1 in blue)**. In this new section, we do a walkthrough of NLP embedding models and discuss how various improvements in NLP embedding models can be explained via implications of CSC based framework.
>
> **Contribution and Utility:**
> We unify the different conjectures such as - distributional hypothesis, non-local-means , etc (which are the basis of different algorithms in literature) into a single mathematical problem of continuous stable-coloring. This problem leads to the formulation of the Global GAL framework. We show that current state-of-the-art algorithms in these fields solve different instantiations of  Local GAL problem which gives a solution that is provably close to the solution of Global GAL problem which. This generic framework highlights that art of embedding model design lies in devising good underlying domain graphs and $f_{agg}$ functions.
>
> Against this backdrop, we can evaluate the existing algorithms and understand them better. See newly added section A.1 in appendix. For example, “Why does BERT perform better than Word2Vec?” -  Using different layers, BERT can be interpreted to be using higher order tokens in the underlying graph. While word2vec uses 1-gram tokens, BERT with L layers uses upto L-gram tokens. Thus, one can expect BERT to perform better as it incorporates more information.
>
> We can also use this framework to seek further research directions. For example,
> “How can we improve an algorithm like word2vec ?“ - Word2vec improvements can come from a variety of considerations such as
> Can we have a better $f_{agg}$ ? One can see AWE as implementing better $f_{agg}$ over word2vec and we see that AWE beats word2vec
> Can we have a better estimation of chosen $f_{agg}$. Right now we only consider sentences of fixed context. This only gives us an estimation of $f_{agg}(N(u))$ as it samples from the neighbourhood. However, can we have better sampling?
> Can we better model the underlying graph? One can see that BERT considers more informative graph and beats Word2vec
>
> **Other details:**
>
> On theorem 4.1:
>
> We have reworded theorem 4.1 and removed the term adjacency matrix.
> For the construction of the word2vec graph, connect each word to every other word. The label of the edge between word $i$ and word $j$ is the co-occurrence count. Then, theorem 4.1 follows from definition 2.1 for an edge-labelled undirected graph.
>
> On other theorems :
>
> While these statements are important results of the paper, we believe that an avid reader will find the details boring as they naturally follow from the formulation. Undoubtedly, the major contribution is in formulation of CSC and related optimization problems.
>
> **Minor questions:**
> 1) neighborhoods in terms of the co-occurrence matrix :  Co-occurrence matrix, in this case, defines the adjacency matrix of the graph . i.e it defines the weighted edges E in G(V,E).

---

### Official Review · Reviewer_JX7R · 2021-11-02

**Correctness:** 3
**Technical Novelty And Significance:** 3
**Empirical Novelty And Significance:** Not applicable
**Recommendation:** 5
**Confidence:** 4

**Main Review:**

This paper provides a novel perspective for understanding unsupervised learning algorithms, which could inspire new research directions. For example, the connection between BERT and stable coloring could help understand the role of different attention layers in BERT. However, the theory of this paper is not proven rigorously and it lacks new method or empirical results.

Comments & Concerns:
 - I find the connection between the original 1-WL algorithm defined for the graph isomorphism test and the global GAL formulation a bit over-stretch. In particular, 1-WL is an iterative algorithm that keeps assigning new colors to unseen neighborhood until the overall node partition in a graph by the color is unchanged; 1-WL does not explicitly push the difference between colors in the current iteration to be similar to that of the next iteration. As a result, if one directly use the GAL in equation 9 as the objective function, then there is a trivial minimum where all nodes have the same embedding; therefore, the authors "proposed" negative sampling as some sort of regulation. However, negative sampling does not have a correspondence in the original 1-WL algorithm.
 - The connection between local GAL and BERT is also over-stretch. Although there is clear analogous between the multiple attention layers with multiple iterations of 1-WL algorithm. In BERT the weights of multi-head attention in different layers are different. Does this mean that the aggregation function in GAL can be different for different layers? It would be helpful if the authors can provide a rigorous derivation of how the proposed setting leads to the original BERT training mechanism, instead of simply saying it is obvious.
 - In theorem 5.1, the authors claim that the $f_{agg}$ is simply an injective hash function that converts multi-sets to an integer; this is not true since if $f_{agg}$ is injective, then new colors could keep being generated forever, and that the stable coloring algorithm would never converge. It is possible that the authors are referring to a special variant of stable coloring algorithm. In that case I think a more rigorous proof would help understanding.
 - In equation 10, the authors use the coloring function $\mathcal{C}$, while in equation 11 the embedding $E$ is used. For consistency it is better to use $E$ for both.
 - Theorem 5.2 shows that L-GAL is sufficient to guarantee good solution for G-GAL. However, it may not be necessary. In fact, in 1-WL, the color assigned in current iteration does not need to be the same as the previous iteration for the algorithm to converge, but rather only the partition needs to be the same. I wonder if the author can show the existing algorithms can directly connect to G-GAL.

Questions:
 - How is the neighborhood $\mathcal{N}$ in section 5.1 defined?
 - What is the definition of $f$ in equation 8?
 - Are you assuming the kernel function $\mathcal{S}$ in equation 10 to be strictly < 1?
 - What doe word-pieces ×N in table 1 mean?
 - How is the co-occurance of unigram and bigram defined?

Typos and Grammars:
 - As mentioned before the graph embedding architectures captures capture...
 - $\mathcal{C}(\mathcal{N} (u)) = C(\mathcal{N} (v)$
 - $\mathcal{C}(\mathcal{N}_l (u)) = C(\mathcal{N}_l (v)$
 - Firstly, describe the construction of the graph for a particular domain and sub-graph induction based on the sample in the data.
 - ??? (missing citations)

**Summary Of The Paper:**

In this paper the authors proposed continuous stable coloring (CSC) as a new framework to unify the understanding of several existing unsupervised learning algorithms, including Word2Vec, BERT, and Node2Vec. The authors show how the original stable coloring algorithm can be understood as optimizing the CSC objective function. The authors also show how to reduce the existing approaches to CSC.

**Summary Of The Review:**

the paper is inspiring but major revision is needed

---

> ### Author Response · Authors · 2021-11-19
> **Clarifications and additional discussion of evidence in support of theory**
>
> Thank you for your comments. There was a general concern among reviewers that there is a lack of experimental evidence to support the theory or its implications. We do not perform experiments because we believe that there is ample evidence in existing literature that supports the implications of our theory. **We have added a new section in appendix (A.1 in blue)**. In this new section, we do a walkthrough of NLP embedding models and discuss how various improvements in NLP embedding models can be explained via implications of CSC based framework.
>
>
> **Main questions :**
>
> There seems to be some confusion about the paper connecting the 1-WL algorithm with the iterative (gradient based) solution of G-GAL/L-GAL. This is not the case. We talk about embeddings as being the stable-colored solutions over underlying structured domain graphs. There is no equivalence between 1-WL and solution to L-GAL in an algorithmic sense. While L-GAL is an optimization problem formulation, 1-WL is an iterative combinatorial algorithm. The relationship between L-GAL and 1-WL is subtle in that the solution to L-GAL is an approximate solution to the CSC ( continuous stable coloring problem) while 1-WL solves the (discrete) table coloring problem. We find that most of the confusions arise due to this mis-connection. Please let us know if we misunderstood your questions.
>
> 1) On connection between 1-WL and L-GAL.
> As mentioned above, we do not make this connection.
>
>
> 1.1) Trivial solution
> Trivial solution will not happen when the $f_{agg}$ is injective.. If $f_{agg}$ is not injective (e.g., mean, sum, median), then trivial solutions will exist for the optimization problem.
> 1.2) On introduction of Negative sampling
> The introduction of negative sampling helps in injecting more information into the system that is not explicitly present in graph structure.
>
> 2) L-GAL and BERT :
> The connection that the reviewer highlights between 1-WL and GNN ( and hence BERT) layers is explored well in literature. However, it is only explored only in context of the question "What architecture should a particular network have w.r.t to two considerations - generalizability and expressivity". This is an orthogonal discussion which, when looked at in the context of our paper, tries to answer the question "what is a good $f_{agg}$?" However, our discussion provides a different view of bert, where the complexity of BERT ( and hence its improvement over Word2vec and AWE) stems from considering a more complex underlying graph structure. While Word2vec only considers 1-grams as nodes, BERT considers $k$-grams where $k$ go from 1 to $L$ where $L$ is the number of layers in BERT. This is shown in the proof of theorem 6.2.
>
> 2.1) Different aggregation function for different layers  in BERT
> There is no concept of layers in L-GAL/G-GAL formulation.
>
>
> 4) On Theorem 5.1 and 5.2
>
> The statement is about the stable-coloring solutions and not the algorithms themselves.
>
> **Minor questions:**
>
> 1) Neighbourhood definition: $N(u) = {v | (u,v) \in E}$ for graph $G(V,E)$.
> 2) What is the definition of f in equation 8? It was a typo. $f$ has been corrected to $E$.
> 3) Co-occurrence of bi-grams and uni-grams : for each time a bi-gram, say u-v,  occurs in the context of uni-gram say w, then this contributes a count of 1 towards the co-occurrence (u-v, w)

---

### Decision · Program_Chairs · 2022-01-20

**Decision:**

Reject

**Comment:**

The paper proposes a new framework to express and analyze embedding methods based on the stable coloring problem. Reviewers highlighted as strengths that the paper provides an interesting perspective for understanding one of the central approaches in NLP, graph learning, and other fields --- and as such could inspire promising research directions. However, reviewers raised concerns regarding the significance of contributions (theoretical insights and analysis, relation to prior work, missing empirical evaluation etc.) as well as the clarity of presentation (also with regard to correctness and scope). All reviewers and the AC agree that the paper is not yet ready for publication at ICLR and would require an additional revision to address the aforementioned issues.